# Polyketides as Secondary Metabolites from the Genus *Aspergillus*

**DOI:** 10.3390/jof9020261

**Published:** 2023-02-15

**Authors:** Xuelian Bai, Yue Sheng, Zhenxing Tang, Jingyi Pan, Shigui Wang, Bin Tang, Ting Zhou, Lu’e Shi, Huawei Zhang

**Affiliations:** 1College of Life and Environmental Sciences, Hangzhou Normal University, Hangzhou 311121, China; 2School of Culinary Arts, Tourism College of Zhejiang, Hangzhou 311231, China; 3School of Pharmaceutical Sciences, Zhejiang University of Technology, Hangzhou 310014, China

**Keywords:** fungus, *Aspergillus*, secondary metabolite, polyketide, bioactivity, therapeutic effect

## Abstract

Polyketides are an important class of structurally diverse natural products derived from a precursor molecule consisting of a chain of alternating ketone and methylene groups. These compounds have attracted the worldwide attention of pharmaceutical researchers since they are endowed with a wide array of biological properties. As one of the most common filamentous fungi in nature, *Aspergillus* spp. is well known as an excellent producer of polyketide compounds with therapeutic potential. By extensive literature search and data analysis, this review comprehensively summarizes *Aspergillus*-derived polyketides for the first time, regarding their occurrences, chemical structures and bioactivities as well as biosynthetic logics.

## 1. Introduction

Polyketides are a highly diverse group of natural products having structurally intriguing carbon skeletons, such as polyphenols, macrolides, polyenes, enediynes, and polyethers [1]. These substances encompass an important source of pharmaceutically relevant molecules, such as antibiotics, immunosuppressants, antiparasitics, cholesterol-lowering, and antitumoral agents [2,3,4,5,6]. Biosynthetically, polyketide motifs are biochemically formed by acetyl-CoA units undergoing a sequence of events catalyzed by polyketide synthases (PKS), a multi-enzyme complex that is highly homologous to fatty acid synthase (FAS) [7].

As one of the ubiquitous fungi in nature, the genus *Aspergillus* has recently received much more attention owing to its great biosynthetic potential of secondary metabolites (SMs) with nutritional, agrochemical and medicinal applications [8]. By the end of 2022, over 3100 *Aspergillus*-derived SMs had been isolated and collected in the Dictionary of Natural Products (DNP) database [9]. Among these substances, as many as 343 polyketide derivatives (**1**–**343**) had been discovered and characterized from *Aspergillus* strains. To enrich our knowledge of these molecules and explore their therapeutic potentials, all aspects are well organized and comprehensively summarized in this review, including their biological sources, structural features, biological properties as well as biosynthetic logic.

## 2. *Aspergillus*-Derived Polyketides as Secondary Metabolites

According to structural features, *Aspergillus*-derived polyketides are grouped into fourteen types, including benzophenone, diphenyl ether, furan and furanone, isocoumarin, lignan, naphthalene, phenolic, polyene, pyran and pyranone, quinone, steroid, meroterpenoid, xanthone and miscellaneous, which are respectively introduced below. Detailed information for these chemicals was summarized in Appendix A.

### 2.1. Benzophenones

Benzophenone derivatives (Figure 1) are a class of ketones formed by the direct connection of one carbonyl with two phenyl groups and play an important role in medicinal chemistry [10]. Under nitrogen-limiting culture conditions, strain *A. nidulans* FGSCA4 was found to produce a novel prenylated benzophenone pre-shamixanthone (**1**), which exerted significant inhibition against lipid accumulation in HepG2 cells without cytotoxic effect and displayed a potent reduction of total cholesterol and triglycerides [11,12]. Two new dichlorinated benzophenones **2** and **3** were purified from *A. terreus* C9408-3 [13], and the later compound is a promising immunosuppressant agent targeting the isomerase cyclophilin A (CyPA) [14]. Three benzophenone analogs (**4**–**6**) obtained from a wetland fungus *A. flavipes* PJ03-11 exhibited stronger α-glucosidase inhibitory activities than acarbose [15]. Bioassay-guided fractionation of the EtOAc extract of one marine sponge-derived strain *A. europaeus* WZXY-SX-4-1 led to the isolation of eight benzophenone derivatives (**7**–**14**), of which compounds **9**, **11**, and **12** showed potent radical scavenging activity against DPPH (2,2-diphenyl-1-picrylhydrazyl) and **8** had strong down-regulation of NF-κB in LPS-induced SW480 cells [16]. Moreover, the putative biosynthetic pathway analysis indicates that endocrocin and emodin were their precursors through consecutive oxidation and methylation (Figure 1).

### 2.2. Diphenyl Ethers

*Aspergillus*-derived diphenyl ethers (**15**–**31**, Figure 2) consist of at least two phenyls connected by one or more oxygen atoms. These aromatic polyketides exhibited excellent potential for therapeutic and industrial applications [17]. Two new rare dibenzo-1,4-dioxins, gibellulins C (**15**) and D (**16**), were produced by genetically modified *A. nidulans* through the deletion of a global regulator *LaeB* [18]. F-9775A (**17**) and F-9775B (**18**), originally isolated from *Paecilomyces carneus,* were detected in the crude extract of *A. nidulans* RMS011 and acted as potent inhibitors of protease K, which could inhibit osteoporosis [19]. Tetraorcinol A (**19**) was a new orcinol tetramer isolated from the fermentation broth of the coral-associated fungus *A. versicolor* LCJ-5-4 and displayed weak DPPH radical-scavenging activity with an IC_50_ value of 67 µM [20]. Besides two chlorinated benzophenones **2** and **3**, three diphenyl ethers (**20**–**22**) were also produced by strain *A. terreus* C9408-3 [13], and compound **20** was shown to be a new endothelin binding inhibitor [21]. Strain *A. flavipes* PJ03-11 also manufactured one new diphenyl ether, 5-hydroxymethylasterric acid (**23**), and seven known analogs (**24**–**30**), of which compound **25** exhibited a stronger inhibitory effect on α-glucosidase than acarbose [15]. Diorcinol (**31**) obtained from the fermentation culture of endophytic *A. flocculus* was found to inhibit the growth of chronic myelogenous leukemia cell line K562 at 30 µM [22].

### 2.3. Furans and Furanones

Furans and furanones are the most polyketides produced by *Aspergillus* spp. and display a broad spectrum of biological properties [23]. Structurally, these substances are classified into two major types, including furans and benzofurans (Figure 3) and furanones and benzofuranones (Figure 4, Figure 5 and Figure 6). 

#### 2.3.1. Furans and Benzofurans

Chemical investigation of one *A. niger* strain from the Caribbean sponge *Hyrtios proteus* led to the discovery of a new furan with a unique carbon skeleton, asperic acid (**32**) [24], which was later reisolated from the strain *A. phoenicis* collected in Saskatchewan (Canada) and exhibited potent cytotoxic activity toward the murine lymphocytic leukemia P388 with an ED_50_ value of 0.18 mug/mL and a variety of human cancer cell lines (pancreas, breast, CNS, lung, colon, and prostate) with GI_50_ values ranged from 1.7 to 2.0 μg/mL [25]. Asperfuranone (**33**) was a novel polyketide consisting of a conjugated alkene chain and a furan subunit produced by A. nidulans by replacing the promoter of the transcription activator with the inducible alcA promoter [26]. A gene cluster containing two fungal PKSs (AN1036.3 and AN1034.3) for the biosynthesis of **33** was first characterized (Figure 2), and its mechanism of action (MOA) showed that this compound exerted an inhibitory effect on A549 cells via blocking cell cycle progression and inducing apoptosis [27]. Two prenylated benzaldehyde derivatives (**34** and **35**) were characterized from the marine-derived fungus *A. glaucus* HB1-19 and showed strong radical-scavenging activity [28]. A new benzofuran polyketide (**36)** was produced by soil fungus *A. terreus* X3 but displayed no antimicrobial effect [29]. Flufuran (**37**) was a typical furan polyketide discovered from *A. flavus* 9643 and shown to inhibit *Phytophthora cinnamomi* at 0.2 mg/mL [30,31]. 3,7-Dihydroxy-1,9-dimethyldibenzo- furan (**38**) originally obtained from a mycobiont of the lichen Lecanora cinereocarnea was found to be produced by an endozoic fungus *A. sydowii* SCSIO 41301 from marine sponge *Phakellia fusca* [32,33]. Asperochratide H (**39**) was a new cytotoxic C_9_ polyketide produced by the deep-sea-derived fungus *A. ochraceus*, and its putative biosynthetic route was proposed in Figure 3 [34]. Asperpentenone A (**40**) possesses a rare cyclopentenone-tetrahydrofuran moiety from strain *Aspergillus* sp. SCSIO 41024 [35]. Asticolorins A–C (**41**–**43**) are toxic metabolites manufactured by strain A. versicolor MRC 638 and were characterized by the novel way in which a mevalonate-derived 3,3-dimethylallyl group was used to link two dibenzofuran moieties [36,37].

#### 2.3.2. Furanones and Benzofuranones

*Aspergillus*-derived furanones and benzofuranones are the most commonly isolated polyketides, including furanones, dihydrofuranones, tetrahydrofuranones and benzofuranones. Interestingly, most of these compounds are aromatic and belong to α-furanone. Penicillic acid (**44**) is one of the important furanone antibiotics used to treat bacterial spot disease [38,39]. Versicolactones A (**45**) and B (**46**) were isomeric furanones produced by a coral-associated fungus *A. versicolor* LCJ-5-4, and compound **46** exhibited pronounced cytotoxicity against human pancreatic cancer cells with an IC_50_ value of 9.4 µM [20,40]. Three acyltetronic acid derivatives (**47**–**49**) were produced by strain *A. niger* ATCC1015 through the activation of the dormant PKS-NRPS gene cluster by expressing the transcription factor *CaaR* [41]. 2-Carboxymethyl-3-hexylmaleic acid anhydride (**50**) was purified from an endozoic fungus *A. tubingensis* OY907 in the Mediterranean marine sponge *Ircinia variabilis* and displayed an inhibitory effect on *Neurospora crassa* [42]. A chemical study of a marine-derived strain *Aspergillus* sp. 16-02-1 resulted in the isolation of eight dihydrofuranone analogs (**51**–**58**) with no potent cytotoxicity against human cancer K562, HL-60, HeLa and BGC-823 cell lines [43]. One new polyketide, asperochrins B (**59**), along with three derivatives (**60**–**62**), was isolated from *A. ochraceus* MA-15. Compounds **60** and **61** displayed selective antibacterial activity against *A. hydrophilia*, *V. anguillarum* and *V. harveyi* with IC_50_ values ranging from 0.5 to 32.0 µg/mL [44].

Aspergones A-D (**63**–**66**) were detected in the fermentation broth of a marine sponge-derived strain *Aspergillus* sp. OUCMDZ-1583 and compounds **63** and **64** showed an inhibitory effect on α-glucosidase with IC_50_ values of 2.36 and 1.65 mM, respectively [45]. Dihydropenicillic acid (**67**) was purified from the endophytic fungus *A. flocculus* [22] but displayed no antimicrobial or cytotoxic activity [46]. Asperochratide F (**68**) was another new C_9_ polyketide from the deep-sea-derived fungus *A. ochraceus* and exerted significant cytotoxic effects on the BV-2 cell line [34]. Gregation B (**69**) was a rare *β*-furanone derived from *A. flavus* in food samples by a qualitative analytical method based on the identification of fungal chemical markers by HPLC-MS [30] and exhibited antibacterial activity against *E. coli* [47]. Avenaciolide (**70**) produced by strain *A. avenaceous* G. Smith displayed an inhibitory effect on the transport of glutamate in rat liver mitochondria [48,49]. Citrifurans A−D (**71**–**74**) was the first heterodimers of azaphilone and furanone from a symbiotic *Aspergillus* strain in the intestines of centipedes and displayed moderate inhibitory activities against LPS-induced NO production in RAW 264.7 macrophages [50]. One year later, two additional new *β*-furanones (**75** and **76**) were obtained from the same strain, and **76** showed significant NO inhibition with an IC_50_ value of 16.0 µM [51].

Asperlactone (**77**) was a new tetrahydrofuranone purified from *A. melleus* CMI 49108 and exhibited superoxide anion inhibition at 30 ± 9% at 10 µM [52,53]. Two new chlorinated polyketides, chlorocarolides A (**78**) and B (**79**), were isolated and characterized from the saltwater culture of *A. ochraceus* [50]. Protulactones A (**80**) possessing unique ring systems was discovered from the marine-derived fungus *Aspergillus* sp. SF-5044 [54]. In addition to compound **47**, tubingenoic anhydride A (**81**) was also produced by strain *A. tubingensis* OY907 and shown to inhibit *Neurospora crassa* growth at 330 µM [42]. Strain *A. ochraceus* MA-15 was found to produce a new C_9_ polyketide asperochrins A (**82**), which showed inhibitory activity against aquatic pathogenic bacterial *Aeromonas hydrophila*, *Vibrio anguillarum*, and *V. harveyi* [52]. Strain *Aspergillus* sp. OUCMDZ-1583-derived aspergiones E (**83**) and F (**84**) displayed α-glucosidase inhibitions [45]. Allahabadolactones A (**85**) and B (**86**) were separated from the endophytic stain *A. allahabadii* BCC45335, and compound **85** displayed moderate cytotoxicity against NCI-H187 and Vero cell lines, and 8**6** exhibited low anti-*B. cereus* effect [55]. Three spiro *β*-furanones, asperones C–E (**87**–**89**), are dimeric polyketides with two distinct skeletons from an unidentified stain *Aspergillus* sp. and compounds **87** and **88** showed significant nitric oxide (NO) inhibition in lipopolysaccharide (LPS)-induced RAW 264.7 macrophage cells with IC_50_ values of 13.2 and 6.0 µM, respectively [51]. Six new C_9_ polyketides (**90**–**95**) were also produced by the marine strain *A. ochraceus*, and compound **94** exerted significant cytotoxic effects on the BV-2 cell line [34].

(+)-Geodin (**96**), originally derived from strain *P. glabrum* AJ117540 was produced by strain *A. terreus* ATCC 20542 and exhibited the activity that stimulates glucose uptake by rat adipocytes [56,57]. Asperetide (**97**) and (5)-3-butyl-7-methoxyphthalide (**98**) were purified from the medicinal plant-derived fungus *Aspergillus* sp. TJ23 [58]. In addition to gibellulins C (**15**) and D (**16**), three porriolide analogs (**99**–**101**) were manufactured by disruption of the global regulator *LaeB* in *A. nidulans* [18] and displayed an inhibitory effect on the root elongation of both lettuce and stone-leek seedlings by 53.3% and 48.5%, respectively [59,60].

### 2.4. Isocoumarins

*Aspergillus*-derived isocoumarins (Figure 7) are a class of phenolic compounds usually containing hydroxyl group(s) and display various pharmacological properties, including antimicrobial, anti-inflammatory, cytotoxic activities and inhibitory effects on serine protease and gamma-secretase [61,62,63]. Chemical investigation of an Indo-Pacific marine sponge-derived *A. ochraceus* afforded a new dihydroisocoumarin, (−)-(*R*)-mellein (**102**), which exhibited a broad spectrum of antifungal and antioomycetes activities [64]. One marine-derived strain *A. ochraceus* MA-15 was shown to produce four isocoumarin derivatives (**103**–**106**), of which compound **106** had inhibitory activity against aquatic pathogenic bacterial *Aeromonas hydrophila*, *Vibrio anguillarum*, and *V. harveyi* [44].

In addition to aspergones A–D (**69**–**72**), five isocoumarins (**107**–**111**) were also obtained from the endozoic strain OUCMDZ-1583 [45], and compounds **107** and **109**–**111** showed α-glucosidase inhibitions with IC_50_ values of 0.027, 1.65, 1.19, and 1.74 mM, respectively, and **107** and **109** exhibited inhibitory activity against the influenza A (H1N1) virus. (3*S*)-5-Hydroxymellein (**112**), originally derived from *Cephalosporium* sp. AL031 was found to be produced by the marine sponge-derived fungus *Aspergillus* sp. SCSIO XWS03F03 [65,66]. Aflatoxins B_1_, B_2_, and G_1_ (**113**–**115**) are a kind of naturally occurring carcinogens frequently detected in secondary metabolites of *A. flavus* [30,67,68]. Compounds **116**–**123** are dihydroisocoumarin derivatives separated from the endophytic strain *A. flocculus* and the marine strain *A. terreus* SCSIO 41008 and displayed no potent cytotoxic effect on chronic myelogenous leukemia cell line K562 [22,69]. Alternariol 9-O-methyl ether (**124**) was isolated from an endophytic strain *A. fumigatus* D but exhibited no antimicrobial activity [70].

### 2.5. Lignans

Lignans mainly exist in plants and have the function of scavenging free radicals and antioxidation [71]. Interestingly, some of these substances had been isolated and characterized from microorganisms, including *Aspergillus* strains (Figure 8). Chemical investigation of the fumaroles-derived strain *A. terreus* C9408-3 afforded four lignan derivatives (**125**–**128**), which compounds **126** and **127** exhibited mild cytotoxic activity, and **128** showed antiplasmodial activity against *Plasmodium falciparum* K1 with an IC_50_ value of 7.9 µg/mL [13,72,73,74]. Three new butenolides (**129**–**131**) together with flavipesin B (**132**) and butyrolactone II (**133**) produced by the fungus *A. flavipes* PJ03-11 displayed stronger α-glucosidase inhibitory activity than acarbose [15]. Microperfuranone (**134**) was a biphenyl furanone polyketide purified from *A. nidulans* [21,75]. Aspergillosis (**135**) and (±)-asperteretal D (**136**) were obtained from cultures of the potato endophytic fungus *A. carneus* L03 and showed moderate antifungal activity against plant pathogens and inhibitory effect on nitric oxide production in lipopolysaccharide-stimulated RAW264.7 cells [76].

### 2.6. Naphthalenes

Naphthalenes, a kind of polycyclic aromatic hydrocarbon composed of two benzene rings sharing two adjacent carbon atoms, are toxic to the liver and nervous system and usually cause cataracts and retinal hemorrhage [77,78]. Six naphthalenes **137**–**142** (Figure 9) were separated from the marine-derived fungus *A. glaucus* but showed no cytotoxicity at 100 µM against the HL-60 and A-549 cell lines [79]. Using heterologous expression in model host *A. nidulans* RJMP1.49, three analogs neosartoricins B-D (**143**–**145**) were biosynthesized and identified [80]. Funalenone (**146**) was produced by an epigenetic regulator gene-deleted strain *A. niger* FGSC A1279 and displayed an inhibitory effect on type I collagenase activity at 170 µM [81]. Two hydroxynaphthalene-2-carboxylate (**147**,**148**) were derived from the marine fungus *A. terreus* SCSIO 41008 and showed weak or no cytotoxic activities toward human glioma U87 cells and glutamate-induced toxicity in HT22 cells [69].

### 2.7. Phenolics

Phenolics are a class of aromatic compounds containing one or more hydroxyl groups and usually act as antioxidants in a number of ways [82]. Orsellinic acid (**149**) and lecanoric acid (**150**, Figure 10) were isolated from A. nidulans RMS011 through co-cultivation with a collection of 58 soil-dwelling actinomycetes. Compound **150** was originally isolated from the lichen *Parmotrema tinctorum* and had a toxic effect on HepG2 and CCF cell lines [83,84]. Bioactivity-guided fractionation of the crude extract of the fungus A. versicolor from a marine sponge *Petrosia* sp. afforded a new aromatic polyketide (**151**), which showed no cytotoxicity against cell lines A-549, SK-OV-3, SK-MEL-2, XF498 or HCT-15 [85]. Seven phenolics (**152**–**158**) from the marine strain A. glaucus HB1-19 exhibited strong radical-scavenging activity [28]. Flavipin (**159**) produced by endophyte A. fumigatus AF3-093A from the brown alga displayed broad-spectrum antimicrobial activity [86]. Porosuphenols A−D (**160**, **161**, **162a** and **162b**) were obtained from the endophytic strain *A. porosus* and possessed a dynamic diene-dione functionality within a flexible carbon chain [87]. Hydroxysydonic acid (**163**) had been isolated from *A. flavus* 9643 and *A. sydowi* and showed NO inhibitory effects in LPS-stimulated BV2 cells [88,89]. A sponge-derived fungus *Aspergillus* sp. F40 was shown to produce a new aliphatic benzoic acid (**164**) with moderate antimicrobial activities [90,91]. Bioactivity-guided isolation and MS-based metabolomics analysis of the endophytic *A. flocculus* resulted in the discovery of three novel phenolics (**165**–**167**) [22]. Eight phenolic polyketides (**168**–**175**) were identified from the marine fungus *A. sydowii* SCSIO 41301, and **172** displayed antimicrobial activity [33,92,93]. Antioxidant agent **176** was the precursor of caffeic acid 3,4-dihydroxyphenethyl ester from the deep-sea fungus *Aspergillus* sp. SCSIO 41024 [35,94].

### 2.8. Polyenes

Polyene polyketides are one kind of important antibiotic which are widely used in the treatment of microbial infections [95]. Structurally, *Aspergillus*-derived polyenes are linear chain molecules (Figure 11). Fumagillin (**177**), discovered from *Aspergillus* sp. in 1949, was shown to be an antiphage agent [96]. Aspinonene (**178**) was a new multifunctional fungal metabolite isolated from the culture broth of *A. ochraceus* FH-A6692 [97]. Compounds **179**–**182** are new C_9_ polyketides and exhibited a weak antitumor effect on K562, HL-60, HeLa, and BGC-823 cell lines but no anti-MRSA activity [58,98]. Aspergones I−M (**183**–**187**) were purified as new polyketides from the strain *Aspergillus* sp. OUCMDZ-1583 and compounds **184** and **185** displayed strong α-glucosidase inhibitions with IC_50_ values of 2.37 and 2.70 mM, respectively [45]. A new antibacterial polyketide (−)palitantin (**188**) was isolated from *A. fumigatiaffnis*, an endophytic fungus on the medicinal plant *Tribulus terestris* and inhibited the growth of multi-resistant clinical isolate of *Enterococcus faecalis* and *Streptococcus pneumoniae* with a MIC value of 64 µg/mL [99].

### 2.9. Pyrans and Pyranones

#### 2.9.1. Pyrans

Recently pyran derivatives received more and more attention due to their wide biological activities, including antibacterial and antifungal activities, and many of them have been developed as commercial antimicrobial agents, such as triadimefon, triadimenol, diniconazole, myclobutanil and bitertanol [100,101]. Azaphilones (**189**–**193**, Figure 12) are a class of highly oxygenated pyrano-quinone bicyclic chemicals from strain *A. niger* ATCC 1015 by activation of a silent PKS gene (*aza*) [102]. Moreover, their biosynthetic pathways were shown to involve the convergent actions of a highly reducing PKS and a non-reducing PKS. Citrinin (**194**) is a pyran mycotoxin produced by several strains of *Aspergillus*, *Penicillium* and *Monascus*. In addition to toxicity, this compound displayed certain anticancer and neuroprotective effects [103]. Five new benzopyran derivatives (**195**–**199**), including two pairs of enantiomers, were purified from the fermentation broth of *A. fumigatus*, an endophytic fungus associated with *Cordyceps Sinensis*. Compounds **195** and **197** exhibited a moderate inhibitory effect on the MV4-11 cell line in vitro with IC_50_ values of 23.95 µM and 32.70 µM, respectively [104]. Two new C_9_ pyran polyketides, asperochratides I (**200**) and J (**201**), were isolated from the deep-sea-derived *A. ochraceus* but showed no cytotoxic, anti-food allergic, anti-H1N1 virus and anti-inflammatory activities [34].

#### 2.9.2. Pyranones

Protulactone B (**202**, Figure 13) was a new α-pyranone polyketide possessing unique ring systems isolated from an EtOAc extract of the marine-derived fungus *A.* sp. SF-5044 [54]. Chaetoquadrin F (**203**) produced by strain *A.* sp. 16-02-1 showed antitumor activity against HeLa cell lines with an inhibitory rate (IR) of 13.5% at 100 μg/mL [43]. In addition to asperochrins A (**82**), five pyranone derivatives (**204**–**208**) were also obtained from strain *A. ochraceus* MA-15 and compounds **205** and **206** displayed inhibitory activity against aquatic pathogens *A. hydrophila*, *V. anguillarum*, and *V. harveyi* [44].

By the heterologous expression of the avirulence gene *ACE1* in *A. oryzae* M-2-3, two new polyenyl-α-pyranones (**209** and **210**) were produced and shown to be not responsible for the observed *ACE1*-mediated avirulence [105]. (+)-Asperlin (**211**) was discovered from an *A. nidulans* mutant, which fused the DNA-binding domain of a transcription factor associated with a silent SM gene cluster with the activation domain of a robust SM transcription factor *AfoA* [106]. Deletion of the epigenetic regulator gene, a histone acetyltransferase in the SAGA/ADA complex, resulted in the production of a novel compound, nigerpyrone (**212**) in *A. niger* FGSC A1279 [107]. Moreover, its biosynthetic pathway was disclosed via gene knockout and complementation experiments (Figure 4). Aspopyrone A (**213**) was produced by an Okinawan plant-derived fungus, *A.* sp. TMPU1623 exhibited a strong inhibitory effect on protein tyrosine phosphatase (PTP) 1B with an IC_50_ value of 6.7 µM [108]. Bioactivity-guided fractionation of the crude extract of an endophytic strain, *A. flocculus*, resulted in the isolation of three pyranone analogs (**214**–**216**) [22]. 4-Hydroxy-3,6-dimethyl-2-pyrone (**217**) and 4-methyl-5,6-dihydropyran-2-one (**218**) were also produced by the marine strain *A. sydowii* SCSIO 41301 [33], and phomapyrone C (**219**) together with compounds **40**, **176** and **215** was purified from strain SCSIO 41024 [35].

#### 2.9.3. Benzopyranones and Naphthopyranones

Aspergchromones A (**220**) and B (**221**), together with noreugenin (**222**, Figure 14), were two new benzopyranones from the marine sponge-derived strain SCSIO XWS03F03 [65]. By deletion of the epigenetic regulator *gcnE* in strain *A. niger* FGSC A1279, two naphthopyranones, aurasperones A (**223**) and Fonsecinone D (**224**) were synthesized, and compound **223** showed a potent inhibitory effect on brine shrimp with an LD_50_ value of 9 ppm [107,109]. In addition to **124**, five naphthopyranone analogs (**225**–**229**) were also produced by the symbiotic strain *A. fumigatus* D but displayed no potent antimicrobial activity [70].

### 2.10. Quinones

Quinones constitute an important class of naturally occurring compounds containing unsaturated cyclic ketone(s) [110]. On the basis of chemical structure, *Aspergillus*-derived quinones (**230**–**277**) could be divided into three types, including anthraquinone, benzoquinone and naphthoquinone, in which the first is the major subgroup [111].

#### 2.10.1. Anthraquinones

Anthraquinones are a group of structurally diverse and biologically active natural products with therapeutic effects [112,113]. Several chemical studies suggested that the marine-derived fungus *A. glaucus* HB1-19 was a versatile producer of anthraquinone polyketides (**230**–**242**, Figure 15), which compounds **230** and **231** displayed potent cytotoxicities against A-549, HL-60, BEL-7402, and P388 cell lines and **241** and **242** had strong inhibitory effects on the receptor tyrosine kinases (RTKs) c-Met, Ron, and c-Src with low-micromolar IC_50_ values [79,114,115]. In addition to the aromatic polyketide **151**, substances **243**–**247** were obtained from the marine strain *A. versicolor*, and **243**, **244**, and **246** exhibited significant cytotoxicity against five human solid tumor cell lines (A-549, SK-OV-3, SK-MEL-2, XF-498, and HCT-15) with IC_50_ values in the range of 0.41–4.61 µg/mL and **243** and **246** also showed excellent antibacterial activity against several clinical Gram-positive strains with MIC values of 0.78–6.25 µg/mL [85]. Sanghaspirodins A (**248**) and B (**249**) were two novel antiproliferative agents from strain *A. nidulans* grown in a chemostat under nitrogen limitation [116]. Two anthraquinones (**250** and **251**) were synthesized by inducing the expression of the silent PKS gene in *A. nidulans* FGSCA4 under a continuous cultivation regime [117]. Compounds **252** and **253** were produced by the fumarole-derived strain *A. terreus* C9408-3 when cultured at 40 °C for 7 days on potato dextrose agar plates [13]. Dermolutein (**254**) and methylemodin (**255**), along with compounds **240** and **256**–**258**, were purified from the EtOAc extract of *A. europaeus* WZXY-SX-4-1 and exerted remarkable down-regulation of NF-κB in LPS-induced SW480 cells [16]. By disruption of a global regulator *LaeB* in *A. nidulans*, a potent aggregation inhibitor asperthecin (**259**) was identified from a mutant by a filter trap assay and electron microscopy [118]. Versiconol B (**260**) together with three analogs (**247**, **261**, **262**) produced by strain *A.* sp. F40 showed weak antimicrobial activity against *S. aureus* and *V. parahaemolyticus* [90]. In addition to the common metabolite **234**, compounds **263**–**267** were detected in the crude extracts of two marine strains *A. sydowii* SCSIO 41301 and *A. terreus* SCSIO 41008 [33,69], and **234**, **264** and **265** exhibited broad inhibitory activities against H1N1 and H3N influenzas. Whereas strain *A. versicolor* HBU-2017-7-derived, two anthraquinones (**268**) and (**269**) showed no antibacterial or cytotoxic activity [119].

#### 2.10.2. Benzoquinones and Naphthoquinones

By HPLC-MS analysis, a toxic benzoquinone spinulosin (**270**, Figure 16) was detected in the SMs of several *A. flavus* strains and displayed effective nematicidal activity against *B. xylophilus* without any plant growth inhibition [30,120,121]. Terreic acid (**271**) produced by strain *A. terreus* ATCC 20542 was a potential anticancer agent with an inhibitory effect on Bruton’s tyrosine kinase [56,122]. Phomaligol A (**272**) and phomaligol A1 (**273**) were two new isomeric benzoquinones discovered from the fermentation culture of *A. flocculus*, and the later possessed a moderate anti-trypanosome activity against *T. brucei* with an MIC of 25 μg/mL [22]. Csypyrone B1 (**274**) was identified as a *csyB* gene product by overexpression under the control of α-amylase promoter in *A. oryzae* M-2-3 [123]. A new naphthoquinone derivative, aspergiodiquinone (**275**), was obtained from a marine-derived *A. glaucus* HB1-19 [28]. From the solid rice medium of marine strain SCSIO XWS03F03, (4*S*)-6-hydroxyisosclerone (**276**) and (-)-regiolone (**277**) were discovered, while the later was shown to be a phytotoxin [65,124].

### 2.11. Steroids

Steroids are cyclopentane polyhydrophenanthrenes and play an important role in life activities [125,126]. Ergosterol (**278**, Figure 17) was isolated and identified from an endophytic strain *A.* sp. TJ23 and exhibited anticancer activities against cell lines B16, MDA-MB-231, 4Tl, HepG2 and LLC with IC_50_ values ranging from 5.13 to 12.3 µM [63]. An ergosterol peroxide (**279**) and campesterol (**280**) were obtained from the fermentation culture of an oyster-derived *A. flocculus* by using modern metabolomics technology [22], and the former displayed an inhibitory effect on the migration of MDA-MB-231 cells at <20 µM [127,128]. An epoxide steroid (**281**) was discovered from the deep-sea strain *A.* sp. SCSIO 41017 was shown to possess moderate activity against cancer cell lines SF-268, MCF-7, HepG-2 and A549 with IC_50_ values of 13.5–18.0 µM [129].

### 2.12. Meroterpenoids

Meroterpenoids as polyketide-terpenoid hybrids are a family of fungal metabolites possessing significant biological activities [130]. However, only a small group of meroterpenoids (**282**–**292**, Figure 18) had been isolated and characterized from *Aspergillus* strains. Terretonin (**282**), produced by a strain of *A. terreus*, had a novel, heavily oxidized 25-carbon skeleton and was presumably derived from the degradation of a triterpene precursor [131]. Co-cultivation of a strain of *A. fumigatus* with the actinomycete *Streptomyces rapamycinicus* afforded the production of two new prenylated polyketides (**283** and **284**) [132]. Parasiticolide A (**285**) was the common SM of two strains of *A. flavus* and *A. parasiticus* IFO 4082 [30,133]. Spiroaspertrione A (**286**) was a novel terpene-polyketide hybrid bearing a unique spiro[bicyclo[3.2.2]nonane-2,1′-cyclohexane] carbocyclic skeleton produced by strain *Aspergillus* TJ23 and performed as an effective potentiator for oxacillin in suppressing MRSA growth by reducing the oxacillin MIC up to 32-fold [134].

Additionally, chemical analysis of the liquid cultures of strain TJ23 resulted in the discovery of two novel terpene-polyketide hybrids (**287** and **288**), of which compound **287** was a potential inhibitor of PBP2a and worked synergistically with the *β*-lactam antibiotics oxacillin and piperacillin against MRSA [135]. Sphaeropsidin A (**289**), along with aspergiloid E (**290**), was obtained from an endophytic fungus *A. porosus* [87] and recently gained interest as a cytotoxic agent, showing selectivity toward melanoma and kidney cancer cell lines with a unique mechanism of action targeting regulatory volume increase [136]. Arugosin C (**291**) was a novel prenylated polyketide produced by a marine-derived fungus, *A. versicolor* HBU-2017-7, but exhibited no inhibitory activity against HCV protease [119,137]. Chlovalicin (**292**) was determined as a new chlorinated meroterpenoid from strain *A. niger* BRF-074 and displayed no cytotoxicity towards the HCT-116 cell line [138].

### 2.13. Xanthones

Xanthones are a class of natural products with hetero-tricyclic structures possessing a variety of biological activities, including antihypertensive, anticonvulsant, antithrombotic, antitumor and so on [139,140,141,142,143]. Two new xanthones (**293** and **294**, Figure 19) were purified from a marine sponge-derived fungus *A. versicolor* [85], and compound **293**, along with its derivative (**295**), was also obtained from strain *A. versicolor* HBU-2017-7 and shown to have significant cytotoxicity [119]. By continuous cultivation for activating silent polyketide BGCs in strain *A. nidulans* FGSCA4, a new prenylated cytotoxic xanthone (**296**) was discovered in its chemostat cultures [144]. Two xanthone dimers (**297** and **298**) originally produced by *A. aculeatus* in 1977 were rediscovered from strains *A.* sp. SCSIO XWS03F03 and *A. aculeatus* IBT 21030 [65,145]. Bioassay-guided fractionation of the crude extract of a soil fungus *A. terreus* X3 resulted in the isolation of penicitrinones A and B (**299** and **300**), which the former showed moderate activity against *B. megaterium* with a MIC value of 1.60 µM [29]. Four prenylated xanthones (**301**–**304**) were separated from the rice medium of the endophytic strain *A.* sp. TJ23 exhibited weak inhibitory activities against the growth of B16, HepG2, and LLC cancer cell lines [58]. Chemical analysis of a marine sponge-derived strain *A. europaeus* WZXY-SX-4-1 afforded six xanthone polyketides (**305**–**310**), of which compounds **305** and **310** exerted excellent down-regulation of NF-κB in LPS-induced SW480 cells [16]. Oxisterigmatocystin I (**311**), along with four analogs (**293**, **312**–**314**), were purified from the culture of a sponge-derived strain *A.* sp. F40 and showed weak antimicrobial activity against *S. aureus* [90]. When cultured under static conditions, strain *A. sydowii* SCSIO 41301 was found to produce two new xanthones (**315** and **316**), which exhibited obvious selective inhibitory activity against H1N1 influenza [33].

### 2.14. Miscellaneous

A number of other bioactive polyketides had been discovered and identified from *Aspergillus* strains. Mevinolin (**317**, Figure 20), along with its analog **318**, was obtained from strain *A. terreus* ATCC 20542 and exhibited a potent competitive inhibitory effect on hydroxymethylglutaryl coenzyme A (HMG CoA) reductase [146]. Aspermytin A (**319**) was a new neurotrophic agent produced by a mussel-derived strain of *Aspergillus* [147]. Three decaline derivatives (**320**–**322**) showed significant cytotoxicity against melanoma cell lines [148,149]. Calbistrin A (**323**), together with its analog (**324**) derived from strain *A. aculeatus* IBT 21030, acted as an excellent antifungal agent, a promoter of nerve growth factor (NGF) production and a cholesterol-lowering substance [150,151]. Two lovastatin analogs (**325** and **326**) were detected in the solid culture of *A. versicolor* SC0156 [152]. Aspergones N-Q (**327**–**330**), along with epoxyquinol (**331**) were separated from the fermentation broth of *A.* sp. OUCMDZ-1583 and displayed strong α-glucosidase inhibitory effects [45]. Salimyxin B (**332**) produced by the endophytic strain *A.* sp. TJ23 showed inhibitory activities against HepG2 with an IC_50_ value of 9.87 µM [58]. Hexylitaconic acid (**333**) was a binary fatty acid originally derived from a marine-derived fungus *Arthrinium* sp., was also produced by the strain of *A. niger* and showed potent antibacterial and antioxidant activities as well as good inhibitory effect on acetylcholinesterase and p53–HDM2 interaction [41,153,154].

A terrein glucoside (**334**) was a new angiogenin secretion inhibitor produced by strain *A.* sp. PF1381 [155]. Bioassay-guided isolation of an extract of *A.* sp. MF6215 led to the discovery of three novel 11-membered macrocyclic biphenyl ether lactones (**335**–**337**), in which compound **335** inhibited the IgE binding to its receptor by an IC_50_ value of 200 µM [156]. By UHPLC-DAD-HRMS and dereplication, aculenes C and D (**338** and **339**) were isolated from a strain of *A. aculeatus* but showed weak antifungal activity [150]. Dehydrocurvularin (**340**) was a new lactone polyketide from strain *A. terreus* ATCC 20542 and acted as a prevalent fungal phytotoxin with heat shock response and immune-modulatory activities and a broad-spectrum inhibitor of various cancer cell lines in vitro [61,157,158]. Aspergones G and H (**341** and **342**) produced by the strain *A.* sp. OUCMDZ-1583 displayed no cytotoxic activity [45]. *A. flavus*-derived terrein (**343**) was a novel suppressor of ABCG2-expressing breast cancer cells MCF-7 cells [13,30].

## 3. Conclusions and Perspectives

In summary, the genus *Aspergillus* is a prolific source of polyketides with diverse chemical structures and a variety of biological activities. Many of these substances or derivatives have therapeutic effects, such as the immunosuppressant agent (**3**), the antioxidant benzaldehydes (**34**,**35**), the α-glucosidase inhibitors (**327**–**330**), etc. Furthermore, the potential to discover novel polyketides from *Aspergillus* strains is still immense since a great number of their BGCs are shown to be inactive or unawakened under traditional culture conditions [159]. With the development and application of bioinformative tools and analytical techniques, more and more *Aspergillus* genomes, as well as functional genes, will be sequenced and annotated. These silent BGCs responsible for the biosynthesis of novel polyketides are being disclosed and activated using new strategies, such as the one strain many compounds (OSMAC) approach and genome mining combined with metabolic engineering [8,160,161]. In addition, the biosynthesis of polyketides from acyl-CoA thioesters is catalyzed by various PKSs, which structures of initiation and condensation domains provide valuable insights into the molecular factors governing starter unit selectivity and chain-length control. A detailed understanding of these PKS structural features controlling polyketide biosynthesis and modification offers a powerful tool for the controlled and rational design of novel polyketides through enzyme engineering. Therefore, more efforts should be made to employ biosynthetic engineering approaches to improve the efficient discovery of novel polyketides from the genus *Aspergillus*.

## Data Availability

Not applicable.

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
