# Peer review of "Polyketides as Secondary Metabolites from the Genus *Aspergillus"

_jof, 2023, doi:10.3390/jof9020261_

Round 1
Reviewer 1 Report
Reviewer comments jof-2127437
The submitted paper provides an extensive overview of all reported Aspergillus-derived polyketides in literature. This review highlights the importance and benefits that Aspergillus spp. provide in the production of these metabolites and the potential these compounds may provide for several applications in numerous fields, such as the medical or industry fields. It likewise provides the chemical structures of these metabolites, which could provide potential future ideas for structural modifications to improve or provide relevant bioactivities. I recommend accepting the manuscript for publication with minor revisions.
Minor revisions
1) Overall biological activities of most metabolites are described. However, they are some metabolites in which only the name, number, or origin are defined but not their activities, which is the most relevant part. I would either try to associate an activity/potential role for these compounds or mention that there is no information available in the literature (either within each metabolite or as an overall remark somewhere in section 2.
a. Such as 18, 47-49, 62, 65-66, 78-80, 91-93, 95, 108, 110, 112, 125, 143-145, 165-175 196-199, 214-216, 218-222, 250-253, 275, 283, 297-299, 301, 307-310
2) Figures 7 onwards are not referred to anywhere in the text. I suggest adding their existence in the corresponding section of the text.
3) I would recommend not adding 2-3 consecutive figures such as figure 13, scheme 10 and figure 15. This is difficult to follow, I would separate them throughout the text.
4) The manuscript has in total 22 figures. I recommended adding the scheme figures as supplementary images. Particularly as only limited metabolites precursors are described, likewise why only a few were selected?. In line with this, perhaps additional information on why it is important to mention these pathways in these limited metabolites.
5) Section 2.2 is all in bold.
6) It is mentioned that metabolite 36 has no antimicrobial effect, then what is its use or role?. Also metabolite 124.
7) The past and present tenses are mixed throughout the manuscript and sometimes is confusing. In line 110, it is mentioned that Flufuran was a typical furan polyketide discovered from A. flavus. Is it not typical anymore?. Line 131, A(45) and B (46) were isomeric furanones. Did they change groups?. Line 152, Gregation B (69) was a rare B-furanone or is?. Line 163, Asperlactone (77) is or was a new tetrahydrofuranone?. Lines 302, 330, 345 have similar situations.
8) Line 130, It should be added that penicillic acid is a furanone.
9) Line 147, metabolite 64 is not in bold.
10) In section 2.12, all species names are not in cursive.
11) In section 2.12, metabolites are not in bold like the rest of the text.
12) Some abbreviations might not be obvious to many readers, such as DPPH (line 63), and PKS (290). I suggest mentioning their full name at their first appearance and looking for others who might also be present.
13) Line 395, metabolite 280 is mentioned twice “In addition to 280, ergosterol peroxide (280) and…”

Author Response
Dear respected reviewer,
Thank for very much for your suggestive comments on our manuscript (jof-2127437). According to your kind suggestions, the original work had been carefully revised, which were highlighted in red. Sincerely hope this updated manuscript would be accepted for publication in Journal of Fungi. Our point-to-point reply is as followings:
Q1: Overall biological activities of most metabolites are described. However, they are some metabolites in which only the name, number, or origin are defined but not their activities, which is the most relevant part. I would either try to associate an activity/potential role for these compounds or mention that there is no information available in the literature (either within each metabolite or as an overall remark somewhere in section 2. Such as 18, 47-49, 62, 65-66, 78-80, 91-93, 95, 108, 110, 112, 125, 143-145, 165-175 196-199, 214-216, 218-222, 250-253, 275, 283, 297-299, 301, 307-310.
Our reply: The present review mainly focuses on Aspergillus-derived polyketides and their biological properties (see Table S1). Certainly, some of these metabolites had been reported to be produced by other organisms before and others were rediscovered later, and shown to exhibit other bioactivities. Therefore, there are too many references related to structure-activity study.
Q2: Figures 7 onwards are not referred to anywhere in the text. I suggest adding their existence in the corresponding section of the text.
Our reply: Done as suggested.
Q3: I would recommend not adding 2-3 consecutive figures such as figure 13, scheme 10 and figure 15. This is difficult to follow, I would separate them throughout the text.
Our reply: Done as suggested. Figure 13 was separated from Scheme 4 and Figure 14.
Q4: The manuscript has in total 22 figures. I recommended adding the scheme figures as supplementary images. Particularly as only limited metabolites precursors are described, likewise why only a few were selected?. In line with this, perhaps additional information on why it is important to mention these pathways in these limited metabolites.
Our reply: On basis of chemical structures, all Aspergillus-derived polyketides were grouped into fourteen types and summarized in 20 figures. Although some biosynthetic pathways of these metabolites had been reported, most of them are putative and not well elucidated. Therefore, only four biosynthetic pathways (see Schemes 1-4) were selected as examples.
Q5: Section 2.2 is all in bold.
Our reply: According to the writing format, the section title or subheading is in italic.
Q6: It is mentioned that metabolite 36 has no antimicrobial effect, then what is its use or role?. Also metabolite 124.
Our reply: These compounds were new metabolites but exhibited no antimicrobial effects.
Q7: The past and present tenses are mixed throughout the manuscript and sometimes is confusing. In line 110, it is mentioned that Flufuran was a typical furan polyketide discovered from A. flavus. Is it not typical anymore?. Line 131, A(45) and B (46) were isomeric furanones. Did they change groups?. Line 152, Gregation B (69) was a rare B-furanone or is?. Line 163, Asperlactone (77) is or was a new tetrahydrofuranone?. Lines 302, 330, 345 have similar situations..
Our reply: The past tense is used in this review since all Aspergillus-derived polyketides were isolated and characterized before.
Q8: Line 130, It should be added that penicillic acid is a furanone.
Our reply: Done as suggested.
Q9: Line 147, metabolite 64 is not in bold.
Our reply: Done as suggested.
Q10: In section 2.12, all species names are not in cursive. In section 2.12, metabolites are not in bold like the rest of the text
Our reply: Done as suggested.
Q11: Some abbreviations might not be obvious to many readers, such as DPPH (line 63), and PKS (290). I suggest mentioning their full name at their first appearance and looking for others who might also be present.
Our reply: Done as suggested.
Q12: Line 395, metabolite 280 is mentioned twice “In addition to 280, ergosterol peroxide (280) and…”
Our reply: Done as suggested.
Thanks again for your valuable time and kind help to improve our work.
Huawei Zhang
Ph.D., professor of microbe natural products chemistry
School of Pharmaceutical Sciences
Zhejiang University of Technology
Hangzhou 310014
China
Reviewer 2 Report
A good review. No suggestions.
Author Response
Dear respected reviewer,
Thanks for your positive comment. According to kind suggestions from other referees, the original work had been improved and carefully revised, which were highlighted in red.
Thanks again for your valuable time and kind help to improve our work.
Huawei Zhang
Ph.D., professor of microbe natural products chemistry
School of Pharmaceutical Sciences
Zhejiang University of Technology
Hangzhou 310014
China
Reviewer 3 Report
The authors present an overview of large group of compounds isolated from the genus Aspergillus. Those compounds are secondary metabolites from polyketide-biosynthetic source.
As a review it is of high importance in its field. Nevertheless, it is not easy to follow in detail each structure and citated literature.
Scheme 2 and scheme 10 (It has to be 4) are very similar to previously publications (see references 26 and 107) and as such they have to be well described in the text included their origin. (and probably a permit is needed)
It looks that reference 34 is not well written.

Author Response
Dear respected reviewer,
Thank for very much for your suggestive comments on our manuscript (jof-2127437). According to your kind suggestions, the original work had been carefully revised, which were highlighted in red. Sincerely hope this updated manuscript would be accepted for publication in Journal of Fungi. Our point-to-point reply is as followings:
Q1: As a review it is of high importance in its field. Nevertheless, it is not easy to follow in detail each structure and citated literature.
Our reply: The typesetting in the original work have been improved.
Q2: Scheme 2 and scheme 10 (It has to be 4) are very similar to previously publications (see references 26 and 107) and as such they have to be well described in the text included their origin. (and probably a permit is needed).
Our reply: Thanks for the reminder. These controversial schemes had been carefully modified. Please check them in new version of manuscript.
Q3: It looks that reference 34 is not well written.
Our reply: Sorry for this error, which had been corrected.
Thanks again for your valuable time and kind help to improve our work.
Huawei Zhang
Ph.D., professor of microbe natural products chemistry
School of Pharmaceutical Sciences
Zhejiang University of Technology
Hangzhou 310014
China
Reviewer 4 Report
The authors present a good contribution for those who need a survey on polyketides in Aspergillus.
There are some recommendations, however, meant to improve consistency in the text, and especially to open possibilities for improved impact of this manuscript.
Please use italics for genus- and species names throughout the text. Please check also gene symbols, but not their products, for being given in italics. Please check also consistency with journal style throughout.
The reviewer appreciates the consequent addition of structural formulae for all relevant componds.
From a chemical point of view, this review is helpful, indeed. However, the reviewer strongly recommends to provide more clues on biological action of individual polyketides. In addition to mention functions in the text, which has been done adequately, this could easily be done in the form of structured tables together with a few keywords and of course the relevant literature. This would allow to provide this valuable information for more compounds and would, in any case, provide readers with the ability to find this information quickly.
It is also recommended to provide a quick look on occurrence of similar, comparable componds elsewhere in the fungal world. This point should certainly not constitute a complete survey, but should allow, by providing adequate literature, to direct readers immediately into the right direction. The reviewer is connvinced that this point will increase interest and scientific impact considerably. This is also of interest, because the genus Aspergillus has close phylogenetic relatives among ascomycetes.
Author Response
Dear respected reviewer,
Thank for very much for your suggestive comments on our manuscript (jof-2127437). According to your kind suggestions, the original work had been carefully revised, which were highlighted in red. Sincerely hope this updated manuscript would be accepted for publication in Journal of Fungi. Our point-to-point reply is as followings:
Q1: Please use italics for genus- and species names throughout the text. Please check also gene symbols, but not their products, for being given in italics. Please check also consistency with journal style throughout.
Our reply: Done as suggested.
Q2: From a chemical point of view, this review is helpful, indeed. However, the reviewer strongly recommends to provide more clues on biological action of individual polyketides. In addition to mention functions in the text, which has been done adequately, this could easily be done in the form of structured tables together with a few keywords and of course the relevant literature. This would allow to provide this valuable information for more compounds and would, in any case, provide readers with the ability to find this information quickly.
Our reply: It is a good idea to improve our work. However, it mainly focused on Aspergillus-derived polyketides and their biological properties (see Supporting Materials). However, most of these metabolites had no report about their biological action owing to lack of potent bioactivity.
Q3: It is also recommended to provide a quick look on occurrence of similar, comparable componds elsewhere in the fungal world. This point should certainly not constitute a complete survey, but should allow, by providing adequate literature, to direct readers immediately into the right direction. The reviewer is connvinced that this point will increase interest and scientific impact considerably. This is also of interest, because the genus Aspergillus has close phylogenetic relatives among ascomycetes.
Our reply: Ascomycetes is the largest phylum of Fungi, with over 64,000 species and a diverse habitat. Aspergillus strains are the most commonly isolated fungi from nature including land and sea. A growing body of chemical study indicates that this genus is a treasure trove of bioactive secondary metabolites besides polyketides described herein. In respect to chemical structure of secondary metabolites, it seems that there is no difference between Aspergillus and other fungal genera, such as aromatic polyketides and pyranones.
Thanks again for your valuable time and kind help to improve our work.
Huawei Zhang
Ph.D., professor of microbe natural products chemistry
School of Pharmaceutical Sciences
Zhejiang University of Technology
Hangzhou 310014
China